# Quantifying Emission Factors and Setting Conditions of Use According to ECHA Chapter R.14 for a Spray Process Designed for Nanocoatings—A Case Study

**DOI:** 10.3390/nano12040596

**Published:** 2022-02-10

**Authors:** Antti Joonas Koivisto, Benedetta Del Secco, Sara Trabucco, Alessia Nicosia, Fabrizio Ravegnani, Marko Altin, Joan Cabellos, Irini Furxhi, Magda Blosi, Anna Costa, Jesús Lopez de Ipiña, Franco Belosi

**Affiliations:** 1Air Pollution Management APM, Mattilanmäki 38, 33610 Tampere, Finland; 2Institute for Atmospheric and Earth System Research (INAR), University of Helsinki, PL 64, FI-00014 Helsinki, Finland; 3ARCHE Consulting, Liefkensstraat 35D, B-9032 Wondelgem, Belgium; 4CNR-ISAC, Institute of Atmospheric Sciences and Climate, National Research Council of Italy, Via Gobetti, 101, 40129 Bologna, Italy; b.delsecco@isac.cnr.it (B.D.S.); s.trabucco@isac.cnr.it (S.T.); a.nicosia@isac.cnr.it (A.N.); f.ravegnani@isac.cnr.it (F.R.); f.belosi@isac.cnr.it (F.B.); 5Witek srl, Via Siena 47, 50142 Firenze, Italy; marko.altin@witekgroup.com; 6Leitat Technological Center, c/de la Innovació 2, Terrassa, 08225 Barcelona, Spain; jcabellos@leitat.org; 7Transgero Limited, Cullinagh, Newcastle West, Co. Limerick, V42 V384 Limerick, Ireland; irini.furxhi@transgero.eu; 8Department of Accounting and Finance, Kemmy Business School, University of Limerick, V94 T9PX Limerick, Ireland; 9ISTEC-CNR, Institute of Science and Technology for Ceramics, CNR, National Research Council, Via Granarolo 64, 48018 Faenza, Italy; magda.blosi@istec.cnr.it (M.B.); anna.costa@istec.cnr.it (A.C.); 10Basque Research and Technology Alliance (BRTA), Consiglio Nazionale delle Ricerche, Parque Tecnológico de Alava, Leonardo Da Vinci 11, 01510 Miñano, Spain; jesus.lopezdeipina@tecnalia.com

**Keywords:** spray coating, emission, NF/FF model, exposure, Conditions of Use (CoU), risk assessment, deposited dose

## Abstract

Spray coatings’ emissions impact to the environmental and occupational exposure were studied in a pilot-plant. Concentrations were measured inside the spray chamber and at the work room in Near-Field (NF) and Far-Field (FF) and mass flows were analyzed using a mechanistic model. The coating was performed in a ventilated chamber by spraying titanium dioxide doped with nitrogen (TiO_2_N) and silver capped by hydroxyethylcellulose (Ag-HEC) nanoparticles (NPs). Process emission rates to workplace, air, and outdoor air were characterized according to process parameters, which were used to assess emission factors. Full-scale production exposure potential was estimated under reasonable worst-case (RWC) conditions. The measured TiO_2_-N and Ag-HEC concentrations were 40.9 TiO_2_-μg/m^3^ and 0.4 Ag-μg/m^3^ at NF (total fraction). Under simulated RWC conditions with precautionary emission rate estimates, the worker’s 95th percentile 8-h exposure was ≤171 TiO_2_ and ≤1.9 Ag-μg/m^3^ (total fraction). Environmental emissions via local ventilation (LEV) exhaust were ca. 35 and 140 mg-NP/g-NP, for TiO_2_-N and Ag-HEC, respectively. Under current situation, the exposure was adequately controlled. However, under full scale production with continuous process workers exposure should be evaluated with personal sampling if recommended occupational exposure levels for nanosized TiO_2_ and Ag are followed for risk management.

## 1. Introduction

Process emissions are a point-of-departure for subsequent exposure and health effects, thus, the quantification of them is foundational for effective risk control solutions [1,2] and communication [3]. When process emissions are characterized according to process parameters it is possible to optimize the process according to exposure and risk. For example, Wang et al. [4] showed that, in plasma cutting of stainless steel when arc current was increased from 35 to 42 and 50 A, it increased Cr^6+^ emissions by ca. 60%. Such knowledge can be used to optimize the process according to production rate and Cr^6+^ emissions. 

Exposure modeling tools designed for occupational nano-safety are mainly relying on qualitative control banding exposure assessment tools [5,6,7,8]. Control banding tools are either based on the emission potential or on the exposure estimates and do not require quantitative concentration levels or emission factors [9]. This makes them less accurate while providing different hazard and exposure outputs when compared with each other and with experimental data [10,11,12]. Inconsistent results limit their applicability in safety decision making and there is a need for quantitative exposure assessment tools that can effectively address the existing uncertainties in a model’s output accuracy and precision [12]. Unlike control banding tools, mechanistic exposure models (e.g., [13,14]) require quantitative information about exposure determinants where the emission source is the most relevant exposure determinant [3]. However, nanomaterial’s (NM) process emissions are rarely reported, which limits their use [15]. 

Commonly used mechanistic models in worker inhalation exposure assessment are single compartment model (fully mixed concentrations) or a Near-Field/Far-Field (NF/FF) model that simulates a concentration gradient near the source [16,17]. The NF/FF model is validated in numerous studies and accepted for regulatory occupational exposure assessment when properly applied [17]. Abattan et al. [16] showed that the NF/FF model predictive performance for solvents was within a factor of 0.3–3.7 times the measured concentrations with 93% of the values between 0.5 and 2. Similar results are obtained for various industrial processes [18]. Functioning models can be used to extrapolate the observed concentrations or exposure levels in one operational condition to another. Such extrapolation is needed when production is scaled up or if operational conditions vary significantly within the company or between different facilities [3]. Especially in the nanotechnology industry, scaling up the exposure measurements is needed because many of the processes are not yet adopted to full production scale and full-scale exposure measurements may not be feasible to perform.

Recently, Koivisto et al. [3] demonstrated with pigment and filler pouring in a paint factory how emission source characterization and a probabilistic NF/FF exposure model can be used to answer common chemical safety questions, such as “Is the process sufficiently safe?” and “What are the most efficient risk mitigation method?”. This methodology can be used to quantify safe Conditions of Use (CoU), which describe the operational conditions and risk management measures that are needed to maintain the exposure under well controlled level [3]. CoU are required in the European Chemicals Agency (ECHA) Chapter R.14 [19] and it is applied for efficient safety communication between workers and manufacturers [3]. According to authors knowledge, CoU are not earlier quantified for occupational processes manufacturing or fabricating NMs [5,6,7,8].

In this study, we demonstrate how the concept by Koivisto et al. [3] can be used to quantify safe CoU for an industrial nano-coating process performed with air spray guns in a ventilated chamber. Setting CoU requires quantification of process specific emission rates, identification of relevant exposure determinants, identification of relevant reference exposure/dose values, setting Reasonable Worst-Case (RWC) operational conditions, and specifying emission control needs. Here is explained how this assessment can be performed in an occupational environment by analyzing measured work-area concentration levels with a probabilistic mass balance model. The work is two-fold; first emission rates are calculated according to process parameters by predicting similar concentration levels as measured, and then, extrapolating the measured concentrations in RWC conditions where the process times for example corresponds to a full production of an 8-h workday. A safety call and setting CoU can be made by comparing the exposure potential with a reference exposure/dose value [20,21]. This method is applicable to any process where emissions are known and when there is a reference value for acceptable exposure/dose level. 

## 2. Methods

This work is based on the measurements reported by Del Secco et al. [22] where details related to materials, processes, work environment, instrumentation, and concentration measurements are reported. A video from the process is available by ASINA project [23]. The concentration measurements were used to characterize the process emissions from polymethyl methacrylate (PMMA), and textile substrates coated with titanium dioxide doped with nitrogen (TiO_2_N) and silver capped by hydroxyethylcellulose (AgHEC) Nano Particles (NPs) using air spray guns. Spray process is known to have a limited transfer efficiency, i.e., a fraction of sprayed product is released to the air [15,24]. Here, we quantified the transfer efficiency for the spraying process that was performed in a ventilated chamber. The fraction of NPs released to spray chamber air are escaped to the process room and further from the room ventilated to outdoor air via local exhaust ventilation (LEV). Emission factors (mg-NP/g-NP; mg of NPs per g of sprayed NPs) were quantified for the coating process by performing measurements in the spray chamber, near the chamber (NF) and in the room (FF). The emission rate from the spray chamber to the room was estimated using a NF/FF model. Then, the model was used to predict how different exposure determinants effect the worker exposure (see exposure determinants form [3]). CoU were estimated by comparing the predicted 8-h Time Weighted Average (TWA) exposure levels under Reasonable Worst-Case (RWC) conditions with the Recommended Exposure Limit (REL) values for TiO_2_ at 300 μg/m^3^; [25]) and Ag at 0.9 μg/m^3^; [26]. Here, the CoU was considered adequate when the 95th percentile of the lognormal distribution of 8-h exposure is below 0.1×REL. According to the human exposure categorization scheme by American Industrial Hygiene Association (AIHA), this limit correspond to exposure category D with “Minimal exposure in the workplace, no effects are expected, based on available information” [20] or highly controlled exposure rating category [21].

### 2.1. Coating Process

The coating machine (Figure 1) is conveyor belt operated and transfers the substrate through a plasma neutralizer to the spray chamber and then to a drying oven. The machine is designed for coating up to 120 cm wide textile and plastic substrates. The plasma neutralizer is optionally used to remove surface charge from the substrate. Spraying is performed inside a ventilated chamber. The fully automated sprayer consists of four spray nozzles and is capable of operating with either one, two, or four nozzles simultaneously. The four nozzles are attached to a mobile element that moves horizontally on the substrate, always at the same height. The spray nozzle operates with 270 normal L/min air flow atomizing the coating suspension, which is fed at a flow rate of 200 mL/min per nozzle. After spraying, the substrate is dried in the drying oven at low temperature. 

The process parameters that were expected to affect the emissions were investigated:
(1)Suspension: ethanol containing 1 wt.% TiO_2_N and water containing 0.01, 0.05 and 0.1 wt.% AgHEC.(2)Substrate: textile and PMMA for TiO_2_N and textile for AgHEC.(3)Number of spray nozzles: 1, 2, and 4 nozzles corresponding to coating suspension flow rates of 200, 400, and 800 mL/min, respectively.

### 2.2. Measurements

Measurements took place in the period 15–18 of February 2021. On 15 February, instruments were set-up and background concentration measurements were carried out with offline particulate matter (PM) samplers. On 16 February, TiO_2_N NPs were sprayed both on PMMA and textile substrates, while on 17 February, AgHEC NPs were sprayed on textiles. On 18 February, additional experiments were carried out (data not shown), and instruments were packed off. During the campaign, measurements were obtained at three different positions: NF, FF, and inside the spray chamber (Figure 1A,B shows NF instrumentation; see also additional photos from [22]).

Size resolved particle concentrations and mass concentrations were measured at NF and FF at heights from 1 to 1.3 m. Details of the instrumentation is given in [22]. The real time NF particle measurement position included:
NF particle mobility size distributions (1/cm^3^) measured by Scanning Mobility Particle Sizer (SMPS) in the range from 10 nm to 1 μm. SMPS scan time was ca 4.5 min with a 1.5 min retrace time.NF particle optical size distributions (1/cm^3^) measured by an optical particles counter (OPC) in size range from 0.3 to 30 μm (in 32 channels) with a time resolution of 6 sec.FF particle optical size distributions (1/cm^3^) measured by another OPC in size range from 0.3 to 30 μm (in 32 channels) with a time resolution of 6 sec.

All instruments were calibrated by the manufacturer prior to the campaign. OPC NF and FF were also intercompared at CNR-ISAC by sampling in parallel aerosol particles at different concentrations. The OPC FF concentration was 1.14 OPC NF in the particle concentration range from ca. 600 to 1600 1/cm^3^ with coefficient of determination R^2^ of 0.962.

OFF-line gravimetric PM samples (total fraction) were taken simultaneously inside the chamber and at NF by collecting particles on filters (PTFE, 1 µm porosity) at 50 L/min flow rate (Bravo H-Plus, TCR Tecora, Cogliate, Italy). Air flow velocities were measured with a hot wire anemometer (Terman ANM-0/B, LSI spa, Milan, Italy).

### 2.3. Combining the NF SMPS and OPC Particle Size Distribution 

The particle size distributions by SMPS and OPC were combined to one d*N*/dLog(*D_p_*) particle number size distribution. The combined particle number size distribution, *N_NF_* (1/cm^3^), was based on the mobility size from 0.010 to 0.74 µm and optical size from 0.87 to 8.69 µm. The number count in overlapping size bins of 0.74 and 0.87 μm were approximated as an average number count counted by SMPS and OPC. The correction due to the soft X-Ray (TSI, model 3087) neutralizer instead of the original one ^241^Am (Grimm Mod. 5522) was applied according to Nicosia et al. [27]. Logarithmic width of the bins were calculated as 10-based logarithm [28]. It was assumed that the mobility and optical particle diameters are the same when using the OPC default refractive index of 1.59 + 0*i* for PSL particles at 632.8 nm. The sampling diffusion losses of SMPS were corrected according to Cheng [29].

### 2.4. Assessment of Test Specific Process Mass Emission Rates from the Spray Chamber to the NF

Test specific mass emission rates from the spray chamber to NF can be estimated from the time resolved NF concentrations measured by SMPS, OPC, and the integrative mass concentration measurement by the gravimetric PM sampler, over all experiments. A combined instrumentation is needed because PM sampler requires long collection times compared to individual tests to reach the limit of quantification. SMPS and OPC measure size resolved number concentrations where the effective density is needed to translate the number to mass concentrations. Mass concentration time series are used to calculate test specific mass concentrations, which further are used to calculate mass emission rates from the spray chamber to NF using a NF/FF model. The assessment consists of following steps:Specify number concentration levels for SMPS and OPC and mass concentration level for PM sampler.Calculate process concentrations where background particles are subtracted for SMPS, OPC, and PM sampler.Calculate effective densities using process particle mass concentrations measured by PM sampler and process particle mass size distributions averaged over PM sampling period.Calculate test specific process particle number concentration averages measured by the SMPS and OPC in NF, *N_P,i_* (1/cm^3^), where *i* is from 1 to 12 according to the test number.Calculate test specific mass concentrations in the NF, *m_P,i_* (μg/m^3^), using *N_p,i_* and the effective densities of both NP’s process emission particles.Develop a model to describe the mass flows in the work environment.Use the model to calculate the emission rates, *G* (mg/min), that reproduce the test specific mass concentrations *m_P,i_* in the NF.Use the model to simulate the processes performed during the PM sampling period and compare the measured and simulated concentrations to evaluate the model goodness.Use the model to identify relevant exposure determinants and quantify CoU.

#### 2.4.1. Process Specific Concentration

Process concentration is solely caused by the process emissions, i.e., the contribution of background concentrations is subtracted. Test specific process concentrations were calculated by subtracting background concentrations from the concentration measured during the test cycle, which in this experiment included four sprayings and four pauses, i.e., averaging time was ca. 40 min per test. Here, the pre-process particle size distribution measured before the spray test was used as a background concentration. The process particle concentrations for test *i*, *N_P,i_* (1/cm^3^), were calculated separately for NF and FF.

Process mass concentration measured by the PM sampler, *m_P,PM_* (μg/m^3^), was calculated by subtracting the PM sampler background concentration measured on day 1 from the TiO_2_N and AgHEC coating process concentrations measured during days 2 and 3, respectively. The *m_P,PM_* was specified over tests 1–6 for TiO_2_N experiments and 7–12 for AgHEC experiments. 

The background concentration was assumed to be constant when calculating the process particle concentrations.

#### 2.4.2. Effective Densities of TiO_2_N and AgHEC Process Particles

The particle density was calculated for both NPs process emissions from the respective NF *N_P,i_* and *m_P,PM_* process concentrations. Effective density takes into account particle porosity and shape [30] and it can be calculated from mass and volume concentrations [31]. First, *N_P,i_* concentrations were averaged over all tests, converted first to volume size distribution and then to a mass distribution *m_P_* (μg/m^3^). The density was adjusted so that the averaged mass concentration over PM sampling period was similar as the *m_P,PM_* mass concentration measured from the NF. The following assumptions were used:Spherical particlesConstant density across the particle size distributionConstant background concentrationHomogenous concentration at the SMPS, OPC, and PM sampling locations at NFMobility and optical diameters are the same

This method calculates an effective density related to particles mobility size, optical size classification, and mass measured by weighting the collected particles by PM sampler (see e.g., [32]). 

#### 2.4.3. Test Specific Mass Concentrations at the NF

The NF mass concentrations for individual tests, *m_P,i_*, were calculated from *N_P,i_* process concentrations and NP’s process respective effective densities. It is assumed that the density of emitted particles is fixed.

#### 2.4.4. A NF/FF Model for the Spray Chamber 

The spray chamber (3.5 × 2.0 × 1.0 m and volume ca. 5.3 m^3^) has two openings for a conveyor belt where the substrate is transferred to the spraying area (Figure 2). The chamber was mechanically ventilated with filtered air extracted from the room, *Q_LEV,in_* (m^3^/min), at a flow rate of 48.6 m^3^/min. The air velocity measured at the spray chamber front screen slit was up to 0.03 m/s (<0.01 m^3^/min) and from the conveyor belt screen to the NF was ca. 0.1 m/s (ca. 1.34 m^3^/min per NF). The flow direction was not possible to define. The spray chamber ventilation flow rate depends also on the ambient pressure and number of spray nozzles air flows. Thus, the flow from conveyor belt screen to the NF volume was assumed to be in balance. 

NPs enter the chamber via atomization of the suspension by the spray nozzles. The NP suspension mass flow, ${\dot{m}}_{s,NP}$ (mg-NP/min) is calculated from the suspension feed rate, ${\dot{m}}_{s}$ (mL/min), and NP concentration *C_NP_* (wt.%). Atomized NPs are transferred to the substrate, ${\dot{m}}_{d,NP}$ (mg-NP/min) and a fraction is released to the chamber air *C_chamber,NP_* (mg-NP/m^3^). The released NPs from the spraying process are mainly removed by LEV and exhausted through a filter to outdoors ${\dot{m}}_{LEV,NP}$ (mg-NP/min). However, a small fraction is leaked into NF from the chamber and conveyor belt cover openings ${\dot{m}}_{r,NP}$ (mg-NP/min). Another fraction is escaped to the pre-heater and oven ${\dot{m}}_{e,NP}$ (mg-NP/min) and deposited onto chamber surfaces ${\dot{m}}_{d,chamber,NP}$ (mg-NP/min). Assuming small NPs escape and surface deposition, i.e., ${\dot{m}}_{d,chamber,NP}\approx {\dot{m}}_{e,NP}\approx 0$ mg-NP/min, the NP mass balance inside the chamber is:(1)$${\dot{m}}_{s,NP}\approx {\dot{m}}_{LEV,NP}+{\dot{m}}_{d,NP}+2\cdot {\dot{m}}_{r,NP}.$$

The particles escaping are assumed to be fully mixed with NF volumes, *V_NF_* (m^3^) in NF1 (spray chamber exit with aerosol instrumentation) and NF2 (spray chamber entrance). A fraction of NF concentrations, *C_NF_*, (μg/m^3^), enters back to the chamber via *Q_NF,in_*, which is assumed to be insignificant if the spray chamber is *Q_LEV,in_* = *Q_LEV,out_*. A fraction escapes to FF by an air mixing between NF and FF, *β* (m^3^/min). Due to NF’ symmetry in entrance and exit of the spray chamber, the NF properties are assumed to be the same.

Assuming that *C_FF_*
≪
*C_NF_*, the NF concentration can be estimated with a NF/FF model (see Model 205-2Box.CE.Gv in [33]. Modellings were performed using a probabilistic NF/FF model (TEAS, Exposure Assessment Solutions, Inc., Morgantown, MI, USA). Table 1 shows the common parameters for the spray chamber model NF1 that is assumed to be symmetric with NF2 (Figure 2) and was used to calculate particle emissions from the spray chamber to the NF1 by using the task specific NF mass concentrations.

#### 2.4.5. Transfer Efficiency and Emission Factors

A transfer efficiency, *P_eff_* (-), describes the fraction that is deposited onto the substrate during spraying. Assuming fully mixed concentrations inside the spray chamber the transfer efficiency is: (2)$$P_{eff}\approx 1-\frac{{\dot{m}}_{LEV,NP}+2\cdot {\dot{m}}_{r,NP}}{{\dot{m}}_{s,NP}}=1-\frac{C_{chamber,NP}\cdot Q_{LEV,NP}+2\cdot {\dot{m}}_{r,NP}}{{\dot{m}}_{s,NP}}.$$

The environmental emission of NPs to outdoor air via LEV exhaust, ${\dot{m}}_{env,NP}$ (mg-NP/min), can be calculated using the suspension feed rate, transfer efficiency and exhaust air filter filtration efficiency, $\varepsilon_{M5}$ (-), as:(3)$${\dot{m}}_{env,NP}=C_{chamber,NP}\cdot Q_{LEV}\cdot \varepsilon_{M5}\approx {\dot{m}}_{s,NP}\cdot \left(1-P_{eff}\right)\cdot \varepsilon_{M5}.$$

According to EN 779:2012 [36] and EN 1822:2019 [37] standards a M5 type filter has an efficiency at 0.4 μm between 40% and 60%. 

Emission factors were defined for NF, *EF_NF, NP_* (mg-NP/g-NP), and LEV *EF_LEV, NP_* (mg-NP/g-NP) as the ratio of the NP release and use as:(4)$$EF_{NF,NP}=\frac{2\cdot {\dot{m}}_{r,NP}}{{\dot{m}}_{s,NP}}$$
(5)$$EF_{LEV,NP}=\frac{{\dot{m}}_{LEV,NP}}{{\dot{m}}_{s,NP}}=\frac{C_{chamber,NP}\cdot Q_{LEV}}{{\dot{m}}_{s,NP}}$$

Here, the LEV filter filtration efficiency is not included in LEV emission.

#### 2.4.6. Particle Emission Rates 

Spray process particle emission rates from the chamber to NF were quantified by setting the emission rate so that the modelled mass concentration during the individual spray test was similar to the respective measured NF mass concentration. The modellings were performed using the spray chamber model, Table 1 parameters, and the *m_P,i_* mass concentrations. The method has the following assumptions:
A NF/FF model describes the dispersion of the emitted particles, i.e.,:
○All mass entering the model volume is created at a source inside the NF volume or from incoming ventilation air;○Particles are fully mixed at all times in NF and FF;○There is limited air exchange between NF and FF volumes;○There are no other particle losses from the room air than the FF ventilation.Background particle concentration is constant.Particle properties are independent of number of spray nozzles and substrate.FF concentration is significantly lower than NF during the process, i.e., NF2 concentration has an insignificant effect on NF1 concentration.Particle emissions from the spray chamber to NF2 are assumed to be the same as in NF1.

### 2.5. Setting Conditions of Use (CoU) for the Spray Process

The CoU was evaluated using emission rates and by simulating the exposure levels under different operational conditions [3]. Effects of the general ventilation, room volume, and air mixing between NF and FF were studied. The effect of LEV was modelled using Model 205-2Box.CE.Gv described by Ganser and Hewett [33]. Setting CoU requires exposure limit values, which are currently not available for NPs. Thus, the evaluation was performed using REL values that may not reflect national regulatory occupational exposure limit values based on the materials CAS number.

### 2.6. Worker Airway Deposition Model

The airway deposition (in head, tracheobronchial, and pulmonary region) for workers in the NF conditions for the 12 spray tests was estimated by the Multi Path Particle Deposition Model (MPPD V3.04; [38]). The combined size distribution from SMPS and OPC (d*N*/dLog(*D_p_*)) and the effective densities were used as inputs for the model. Additionally, the following parameters were defined in the model:
The human Yeh/Schum Symmetric lung model [39].Breathing parameters for adult males were selected to reflect moderate worker activity: 3300 mL for functional residual capacity, 50 mL for extrathoracic volume, 20 breaths/minute for breathing frequency, and 1100 mL for tidal volume. This corresponds to an inhalation rate of 1.36 m^3^/h, which is between light (0.6 m^3^/h) to moderate (1.7 m^3^/h) activity for a 81 kg male between the ages 35 and 64 [40].A polydisperse particle distribution for particle diameter as CMD (count median distribution) and the calculated GSD.Aspect ratio was set to “1” (assuming spherical particles).The oronasal (normal augmenter) assuming most humans combine nose and mouth breathing.The “inhalability adjustment” was “off” given that this is only relevant for particle sizes larger than about 8 μm for humans.Up-right body orientation was selected.Test specific mass concentrations, *m_P,i_*, as input for “Aerosol concentration”.Constant exposure conditions and default exposure values (i.e., for acceleration of gravity, breathing frequency, tidal volume, inspiration fraction, and pause fraction).

Deposited dose of TiO_2_N NPs was compared with the no significant dose level of 300 µg/day for particle overload, chronic inflammation, and cell proliferation and with the no significant dose level of 44 µg/day for tumor incidence [41].

## 3. Results

In total, 12 spray tests were conducted. A single test cycle consisted of four spraying events where a single spray includes spraying from 1.5 to 3 min and ca. 8-min pause. Between the tests, a 14–28 min break occurs to prepare for a new test. The overall cycle including a break is ca. 60 min. A single test cycle is demonstrated in Figure 3, and the individual spray times are presented in Appendix A. Appendix A, shows an example how the measurement cycle in Test 1 is related to the measurements. The experimental variables for each test are shown in Table 2 and the experiment duration defined as the PM sampler sampling duration and spraying times are shown in Appendix A.

### 3.1. PM Sampler Mass Concentrations

Background concentration measured on day 1 was 28 ± 3 and 19 ± 2 μg/m^3^ at NF and inside spray chamber, respectively, when measured without chamber ventilation (Appendix A). According to Inductively Coupled Plasma - Optical Emission Spectrometer (ICP-OES) analysis, background concentration did not contain Ti or Ag at detectable quantities.

The concentration during TiO_2_N spraying processes on day 2 was 120 ± 2 and 1217 ± 1 μg/m^3^ at NF and inside the spray chamber, respectively (Appendix A). Process specific concentrations with subtracted background concentrations were 92 and 1180 μg/m^3^, respectively. TiO_2_ concentration calculated from Ti concentration and Ti and O elemental masses was 40.9 and 776.7 TiO_2_-μg/m^3^, respectively, indicating that 44% and 65% of the NF and chamber concentrations was TiO_2_, respectively. Note that nitrate doping is not included in the mass concentrations.

The concentration during AgHEC spraying processes on day 3 was 65 ± 2 and 191 ± 2 μg/m^3^ at NF and inside the spray chamber, respectively (Appendix A). Process specific concentrations with subtracted background concentrations were 37 and 178 μg/m^3^, respectively. Ag concentration was 0.4 and 13.2 Ag-μg/m^3^, respectively, indicating that 1.1% and 8% of the NF and chamber concentrations was Ag, respectively. Note that HEC doping is not included in the mass concentrations.

### 3.2. Particle Number Concentrations

Figure 4 shows an overview of the temporal trend of the particle number concentrations measured by OPCs in NF and FF for TiO_2_N and AgHEC spray tests. The background concentration was between 250 and 500 1/cm^3^ on day 1 and ca. 100 1/cm^3^ on day 2 until ca. 2:00 PM. The background concentration increased to ca. 300 1/cm^3^ between 2:00 and 3:30 PM. This was caused by a combustion engine exhaust outside the warehouse. Ultrafine particles released from the combustion engine are not expected to significantly affect the background mass concentration.

Test specific process particle number size distributions *N_P,i_* are shown in Figure 5 and the values are given in Appendix A. The background concentration is occasionally higher than the measured during the process resulting in negative concentration values in some size channels (Figure 5). Total process number concentrations varied from 516 to 8371 1/cm^3^ for TiO_2_N tests and from 38 to 3708 1/cm^3^ for AgHEC tests (Table 2). The process number concentration *N_P_* averaged over TiO_2_ tests 1–6 was 4740 1/cm^3^ and for AgHEC averaged over tests 7–12 was 2020 1/cm^3^.

The FF particle number concentrations increased slightly during some spraying tests (Figure 5). The FF particle concentrations and size distributions measured by the OPC are shown in Appendix A. Total process number concentrations in FF varied from −9 to 81 1/cm^3^ for TiO_2_N tests and from −4 to 58 1/cm^3^ for AgHEC tests. The FF total mass concentrations were mainly negative, which was caused by the higher background particle concentrations in the size larger than ca. 2 μm (Appendix A). The physical interpretation of negative concentrations is that the background concentration was not constant, and the process concentration was below limit of quantification (not specified).

### 3.3. TiO_2_N and AgHEC Process Particles Effective Densities

TiO_2_N tests 1–6 and AgHEC tests 7–12 average *N_P,i_* concentrations are shown with black solid lines in Figure 5A and Figure 5B, respectively. Those were used to calculate densities. The average mass concentration calculated from SMPS and OPS measurements is 92.4 μg/m^3^ for TiO_2_N experiments 1–6 at density of 2.1 g/cm^3^ and 36.9 μg/m^3^ for AgHEC experiments 7–12 at density of 6.5 g/cm^3^. The calculated average mass concentrations, *m_p_*, are shown in Figure 5 (black dashed lines). The elemental densities of TiO_2_, Ag, and HEC are 4.23, 10.49, and ca. 0.8 g/cm^3^, respectively. Composition and density of the other process emission particles were not evaluated.

### 3.4. Task Specific Mass Concentrations

Task specific mass concentrations, *m_p,i_*, were calculated from task specific average number distributions (Appendix A) using TiO_2_N and AgHEC process particle effective densities. The NF mass concentrations varied in TiO_2_N tests 1–6 from 25.1 to 202.6 μg/m^3^ and in AgHEC tests 7–12 from −4.9 to 71.3 μg/m^3^ (Table 2 and Appendix A). The negative mass concentration (test 7) means background concentration was higher than the concentration measured during the process.

### 3.5. Emission Rates

Modelled task specific mass concentrations were simulated using the NF/FF model. Parametrization of the work environment (Table 1) was the same for all simulations and the task specific process times are given in Appendix A. Because the PM sampler is an integrative sampler it cannot distinguish the individual breaks between the tests and the break before and after the first and last spray test. Thus, the sum of the break times, 111 min for TiO_2_N experiments and 115 min for AgHEC experiments, were divided equally with the individual tests, resulting to 18.5 min break for a TiO_2_N test and 19 min break for an AgHEC test. 

Individual simulation results and deviation from the *m_P,i_* concentrations are shown in Appendix A, and an example for test 1 is shown in Figure 3. The ratio of the measured mass concentration *m_P,i_* and the respective modelled mass concentration varied in TiO_2_N tests 1–6 from 97% to 100% and in AgHEC tests 7–12 from 94% to 102%. The emission rates varied from 0.9 to 5.9 mg/min in TiO_2_N and from 0.8 to 2.6 mg/min for AgHEC experiments (Table 2).

### 3.6. Model Comparison with the Measurements

The experiments were simulated 10,000 times using Table 1 parametrization with the mass emission rates quantified for individual tasks as total process emissions and NPs (Table 2) and the testing times presented in Appendix A. The number of simulations was selected so that the results variation was insignificant when simulation was repeated by the same scenario. Task 7 emission rate was set to 0 mg/min. Simulation time was the same as the PM sampler sampling time.

Reports are given in Appendix A, showing individual model parametrization, exposure distributions, exposure statistics, and job sensitivity analyses for NF and FF concentrations. Simulated random day mass concentration time series in the NF (Figure 6) reproduced similar concentration peaks as measured particle number concentrations shown in Figure 4 for TiO_2_N and AgHEC experiments, except test 7. The measured geometric mean NF concentration for TiO_2_N experiment was 94 μg/m^3^ (GSD 1.111) and for AgHEC experiment was 39 μg/m^3^ (GSD 1.106). Emission rates were adjusted to reproduce test specific mass concentrations (Appendix A). Test specific emission rates were used to reproduce the full day of NF measurements (Appendix A). The ratio of the measured mass concentration *m_P,PM_* and the respective modelled mass concentration was 98% for the TiO_2_N experiment and 94% for the AgHEC experiment, i.e., the modelled concentration was slightly overestimated in both cases. 

FF concentration is caused by mixing concentrations from NF to FF. The average geometric mean FF concentration caused by NF1 was 4 μg/m^3^ (GSD 1.452) for TiO_2_N experiment and 1 μg/m^3^ (GSD 1.444) for AgHEC experiment. Due to the spray chamber symmetry, it is assumed that the entrance increases the FF concentration in a similar amount, while the FF concentration is doubled. This leads to average FF concentrations of 8 and 2 μg/m^3^ for TiO_2_N and AgHEC experiments, respectively. It is good to note that this assumption violates mass balance principles, as noted earlier. However, the effect is not significant. For example, if the whole spray chamber would be enclosed with NF volume having a triangular 3, 6, and 16 m^3^ as minimum, mode and maximum volumes and emissions from both sides would be the same (i.e., the generation rate to the NF volume would be doubled), the average FF concentration in TiO_2_N experiments would be 8 μg/m^3^. Thus, the assumption to double the FF concentration when calculated from single spray chamber side emissions is reasonable.

The emission rate quantification contains assumptions related to the mass balance listed in Table 1 and Section 2.4. The ranges given in Table 1 result to a GSD of up to 1.23 in the concentration simulations of individual tests (Appendix A).

Mixing of concentrations in the NF volume is one critical uncertainty factor, i.e., how homogenously the concentrations are dispersed in the NF. This can be measured using multiple on-line measurements, e.g., diffusion chargers. Another critical uncertainty factor is the flow balance between the spray chamber and NF. It was not known if the chamber was under negative or positive pressure compared to NF. Assuming that the operational conditions can lead to a flow from the spray chamber to the NF, the emission rate increases according to the chamber concentration and volume flow rate. Here, the volume flow rate was below 1.34 m^3^/min and the average chamber concentrations were 1180 and 178 μg/m^3^, respectively, for TiO_2_N and AgHEC processes. On average, this would correspond to a total emission rate of 1.6 and 0.24 mg/min, respectively, for TiO_2_N and AgHEC processes, or, respectively, 60% and 16% of the average emission rate. It is also unclear if emissions from the spray chamber entrance to NF2 can impact on the NF1 concentrations at significant level.

Based on a qualitative emission rate uncertainty assessment, a reasonable range for the emission rate was set as a linear distribution, where the lower limit is the quantified emission rate and the upper limit is twice the emission rate, i.e., ${\dot{m}}_{r}$ ≤ *G* ≤ $2\cdot {\dot{m}}_{r}$. This represents a precautionary parametrization to avoid exposure underestimation in NF and FF.

### 3.7. Transfer Efficiency and Environmental Emissions

Sprayed suspension volumes, mass flows of TiO_2_N and AgHEC NPs to LEV exhaust and room air, calculated transfer efficiencies and emission factors to room and environment are given in Appendix A. The chamber particle number and mass concentrations were too high for optical counting due to particles coincidence counting (data not shown; [42]), which is the reason why the task specific mass flows and transfer efficiencies cannot be specified. Assuming that the concentrations are fully mixed inside the spray chamber, the mass flow of NPs by LEV exhaust was on average 37.3 mg-TiO_2_/min and 0.7 mg-Ag/min when calculated from the LEV volume flow and the spray chamber mass concentration measured by the PM sampler. According to Equation (2), the transfer efficiencies were on average 97.6% and 98.9% for TiO_2_N and AgHEC spraying processes, respectively. Here, the transfer efficiency includes chamber wall deposit, which was assumed to be insignificant. 

Emission factors for NF varied from 0.7 to 3.2 mg-TiO_2_/g-TiO_2_N and from 0.2 to 0.9 mg-Ag/g-AgHEC (Appendix A). The emission factor for LEV (as average) was for TiO_2_N processes 22.6 mg-TiO_2_/g-TiO_2_N and for AgHEC processes 11.1 mg-Ag/g-AgHEC (Appendix A).

### 3.8. Quantifying CoU

Current REL values given as respirable fraction are 300 for TiO_2_ [25] and 0.9 μg/m^3^ for Ag [26] as an 8-h TWA. The predicted exposure risk is considered highly-controlled when a 95th percentile of the exposure concentration distribution is below 10% of the occupational exposure limit value [21,43]. Here, the REL values were used because legally binding exposure limit values do not exist for nanosized TiO_2_ or Ag particles.

The worker is rarely at the NF because the process is automated and controlled from FF. Under RWC conditions, the worker is expected to spend, during the 8-h work shift, 10% at the NF and 90% at FF. It is assumed that the RWC process time in the 8-h work shift is 6-h, divided into two 3-h continuous spray events. The RWC exposure in this worker time use is called weighted exposure. Precautionary emission rates were used assuming linear distribution between ${\dot{m}}_{r}$ ≤ *G* ≤ $2\cdot {\dot{m}}_{r}$, i.e., the upper limit for the emission rate is twice the emission rate from the spray chamber to the NF.

Under experimental conditions (Table 2), the NF NP concentrations were up to 80 TiO_2_-μg/m^3^ and 0.78 Ag-μg/m^3^ when assuming that 44% and 1.1% of the total concentrations contain TiO_2_ and Ag, respectively. 

Under RWC conditions, the NF 95th percentile NP concentrations were up to 924 TiO_2_-μg/m^3^ and 10.6 Ag-μg/m^3^, which is ca. 12 and 14 times higher than during the experiments (Table 3 and Appendix A). Under RWC conditions, the weighted 95th percentile exposure levels range from 26 to 171 TiO_2_-μg/m^3^ and from 0.6 to 1.9 Ag-μg/m^3^ during an 8-h work shift process. 

Under experimental conditions, the NF NP concentrations were up to 27% and 78% of the respective REL for TiO_2_ and Ag, respectively. Under RWC conditions, the NF NP concentrations exceeded up to ca. 3 and 10 times the RELs for TiO_2_ and Ag, respectively. The RWC weighted exposure was up to ca. 0.6 and 1.9 times the REL for TiO_2_ and Ag, respectively. Note that the RWC concentrations were simulated using the total fractions while REL values are given as respirable fractions. Here, the mass fraction of particles below 4 μm was ca. 90% from the measured particle mass size distribution (Figure 5A).

#### 3.8.1. Sensitivity Analysis 

Test 12 RWC conditions were varied to see the effect of general ventilation, room volume, NF volume, and the air mixing flow rate between NF and FF (*β*) to the NF and FF concentration levels and the weighted exposure (Appendix A). Halving the general ventilation rate, room volume or NF volume increases the NF concentration only up to 5%. However, halving the general ventilation rate or room volume, doubles the FF concentration, thus increasing the weighted exposure by 40%. Halving the *β,* doubles the NF concentration but the FF concentration remains at similar level, increasing the weighted exposure by factor of 1.5. Under RWC conditions, where room volume, NF volume, general ventilation, and *β* are halved (Appendix A), the NF and FF concentrations increase by a factor of 2.0 times and 4.0, respectively, which increases weighted exposure by a factor of 2.9.

#### 3.8.2. CoU under RWC Conditions

To meet the condition of 0.1 times REL value of the 95th percentile of the exposure distribution for highly controlled exposure [21], the exposure under RWC conditions should be reduced by a factor of 19 times to achieve compliance with AgHEC coating process. If LEV exhaust flow rate increased by 20 m^3^/min, it would extract the replacement air from the spray chamber inlet and exit at 10 m^3^/min each. This would act as an LEV for the NF. Assuming that NF LEV would reduce the emissions from the spray chamber to room by 80%, the NF and FF 95th percentile concentrations under test 12 continuous spraying process would be 0.56 and 0.032 μg-Ag/m^3^, respectively (Appendix A). The weighted 95th percentile exposure would be 0.084 μg-Ag/m^3^, which is below 0.1 time the Ag-REL value. Additional measurements should be conducted to ensure 80% reduction in emissions to the room. 

### 3.9. Process Parameters Effect on the NF Concentration and Emissions

The NF particle number concentration did not increase systematically with increasing the number of nozzles for TiO_2_N experiments or with increasing AgHEC concentration of the suspension for AgHEC experiments (Table 2; Appendix A). In mass concentration, there was a systematic increase in NF mass concentration with process parameters increasing the spraying process NP mass flow (Table 2; Appendix A). The relation between process parameters and NF concentrations can be used to estimate how much process parameter effect on NF exposure level (Appendix A). However, more relevant for exposure assessment is to understand how much emissions are changed by different process parameters (Appendix A). Such relations can be used to optimize the process according to the worker exposure.

### 3.10. Modeled Deposited Dose in Respiratory Tract in NF Conditions

Deposition fractions and dose rates during inhalation were calculated for each test (Appendix A). Total average deposition fraction was 0.30 (0.08) for TiO_2_N experiments and 0.44 (0.14) for AgHEC experiments, where brackets show standard deviation. The deposited dose rates were 0.20 (0.15) for TiO_2_N and 0.16 (0.12) μg/min for AgHEC experiments. In TiO_2_N experiments, the particles were deposited 10% (6%) in head airways, 7.6% (0.6%) tracheo-bronchial, and 13% (1.5%) pulmonary regions. In AgHEC experiments, the particles were deposited 17% (15%) in head airways, 10% (1%) tracheo-bronchial, and 16% (2%) pulmonary regions. For the same particles, the variation in deposition fractions between the experiments was caused by the differences in the particle size values approximated using a log-Normal distribution. Deposition rates were calculated for individual tests, which showed also variation between the tests.

Under RWC conditions, the total weighted deposited mass (µg/day) in each of the airway regions was also calculated for each test. (Appendix A). Deposited mass in the pulmonary region was 9.0 (µg/day) (7.3) for TiO_2_N experiments and 6.1 (µg/day) (2.7) for the AgHEC experiments.

## 4. Discussion

Setting CoU for an industrial process relies on regulatory limit values such as occupational exposure limit values, which are not available for NMs. Current proposed occupational exposure levels vary for nano-TiO_2_ from 0.8 to 5000 μg/m^3^ and for nano-Ag from 0.098 to 10 μg/m^3^ when given in different size fractions and specified under different experimental conditions [44,45,46]. In this exercise, it was decided to use the NIOSH’s RELs because they are well recognized and provide confidence for use. Since there are no legally binding limit values for NMs, it is not possible to specify regulatory accepted CoU for this process. 

Understanding environmental and occupational risks associated to the spray coating process requires quantification of exposure determinants such as transfer efficiency, and emission rates from chamber to the room and outdoors via LEV system. Understanding workers’ exposure and risk requires additional information about the dispersion of particles in work atmosphere and worker exposure times. This can be obtained by simulating exposures in RWC exposure scenarios or under realistic conditions when the exposure determinants and process specific emissions are known.

In the framework of the Directive 98/24/EC on chemical agents at work, worker exposure by inhalation is traditionally quantified by measuring personal breathing zone concentration across the work shift [31]. However, it is not always obvious if the measurement scenario is representing RWC exposure scenario. Moreover, low exposure limit values bring additional challenges, such as the Ag-REL of 0.9 μg/m^3^. Probabilistic exposure models can be used to predict exposures in all work shift combinations and RWC conditions typically without issues with limit of quantification related to personal sampling [3]. In this case, the measured scenario would be classified *well controlled* for TiO_2_ coating process (95th percentile of the exposure distribution is <0.5 × REL; [21]) and *controlled* for Ag coating process (95th percentile of the exposure distribution is ≤REL; [21]) based on NF concentrations. However, in a continuous coating process using four nozzles, the situation changes to *poorly controlled* because of higher material uses per work shift and due to environmental conditions favoring higher concentration levels (Table 2 and Table 3). 

As a precautionary assumption, all particles can be assumed to consist of NPs. Here, it was found that total NF concentration during TiO_2_N and AgHEC spraying processes contains 44% and 1.1% TiO_2_ and Ag, respectively. Thus, the elemental analysis of collected particles significantly improve the exposure estimates.

Transfer efficiency can be used to calculate environmental emissions according to the material use. For example, TiO_2_N mass flow in coating process is 1.65 g-TiO_2_N/min per nozzle and the transfer efficiency is ca. 97.6%. Thus, emissions to air are ca. 40 mg/min where ca. 6% is released to workplace air from spray chamber inlet and exit and 94% are exhausted via the LEV. The LEV filter filtration efficiency was on average 50%. Thus, the environmental emissions via LEV exhaust are ca. 20 mg/min when assuming LEV exhaust tube losses insignificant. In a six-hour coating process with one nozzle, would be emitted ca. 3600 mg-TiO_2_ particles outdoor air and filter TiO_2_ loading would increase a similar amount. As another example, assuming that factory is consuming 10 kg TiO_2_N NPs per year by using this coating machine, the environmental TiO_2_ emission to air would be ca. 110 g when calculated using an emission factor of 22.6 mg-TiO_2_/g-TiO_2_N and 50% filtration efficiency in LEV. This concept can be coupled directly with environmental exposure models (e.g., [47]).

Process emissions from the spray chamber to the NF can be quantified according to the process parametrizations. In our case, these are substrate (Textile/PMMA), coating suspension type, AgHEC concentration, and number of nozzles (also suspension flow rate). Changing the flow rate from 200 to 400 mL/min increases the NF concentration on average 1.7 times (SD = 0.3) and for TiO_2_N coating process PMMA substrate caused 1.4-, 1.2-, and 1.1-times higher NF mass concentrations than textile at flow rates of 200, 400, and 800 mL/min, respectively (Table 2 and Appendix A).

Thinking about a future sustainable smart manufacturing, process parameter specific emissions and concentration levels can be used by digital technologies, such as Digital Twins, as input data to predict in real time environmental emissions and workers’ exposures, and to optimize the sustainability of the manufacturing process itself [48]. 

### 4.1. Methodological Uncertainties Related to Effective Density

The particle effective density defined from the gravimetric samples and particle number size distributions include several assumptions: (i) spherical particles, (ii) identical mobility and optical size, (iii) constant agglomerates density over the measured particle size range, (iv) pre-process background concentration remains the same during the process. Spherical particle assumption is an intrinsic property of the effective density. 

Mobility and optical size of TiO_2_ and Ag particles are not the same because the optical size also depends on the particle refractive index and shape [49,50,51]. The refractive index and extinction coefficient at 632.8 nm are for TiO_2_ (rutile) 2.8736 + 0*i* and for elemental Ag 0.056 + 4.27*i*, respectively. These vary significantly from the coefficient used by the OPC. For example, for highly agglomerated TiO_2_ NPs [52], the optical diameter was 90 nm + 0.14 times the mobility diameter (Appendix A). However, here, TiO_2_ particles were spherical, doped with nitrate and the relation cannot be used in this case. Agglomerate density reduces with increasing agglomerate size when the primary particle size remains the same that is shown, e.g., for soot agglomerates [53,54]. This assumption overestimates the large particles mass fraction and underestimates small particles mass fractions. The background concentration assumption is expected to cause <10% error on the process particle mass concentration calculation.

### 4.2. Model Uncertainties

Major uncertainties in the model are related to air flow balance between the spray chamber and the NF (*Q_NF,SC_*), how well the concentration measurements represent the NF average concentrations level, and the air mixing flow rate between NF and FF. Here, the particles were assumed to diffuse from the spray chamber, i.e., *Q_NF,SC_* = 0 m^3^/min, but if the LEV flows are not in balance it will change the emissions proportional to the *Q_NF,SC_*. Concentration gradients in the NF volume can cause high under- or overestimation in emission rates. Concentration mapping requires simultaneous measurements with multiple instruments that were not available in this measurement campaign. Here, concentration measurements were expected to be representative for the mean NF concentration level because there were many small openings for particles to leak to the NF forming a local fugitive emission source rather than point source. Air mixing flow rate between NF and FF dilutes the NF concentrations. If dilution is overestimated, i.e., *β* is higher than true value, then the emission rate is overestimated. Here, the measured random air flow speed was up to 1.8 m/min (lower detection limit), which is below the typical random air speed at schools and offices of 2.5 m/min [55]. Modelled concentrations showed small GSD because the parametrization was made according to the best knowledge and it does not account all identified uncertainties, which were included subjectively by increasing the emission rate range.

### 4.3. Deposited Dose

In occupational settings, deposition in the airways depend on the aerosol properties (particle size distribution, particle concentration, density, etc.) that are highly determined by the exposure scenario conditions, such as spraying time and worker location. If the particle size distribution shape and breathing conditions are the same during the work shift, deposition fractions can be used to estimate the deposited dose in airways under different exposure conditions when concentration and exposure duration, i.e., volume of inhaled air, are known. 

The clearance mechanism present in the different parts of the airway will determine the final fate and uptake of deposited particles in the organism [56,57]. In the head and the tracheobronchial regions, most of the materials will be relatively rapidly eliminated either by expulsion or by translocation into the gastrointestinal tract (head region) or cleared by the mucocyliary system and translocated to the gastrointestinal tract (TB region; [56]). In both cases, a small fraction would be subjected to a more prolonged retention. 

In the pulmonary region, higher retention rates may be assumed than for the head and tracheobronchial region. However, this will depend among other factors on the clearance mechanism in the alveolar region (e.g., alveolar macrophage clearance) or transformation (e.g., dissolution in the lung lining fluid) and translocation mechanisms. 

Process parameter specific emissions, aerosol concentration levels and calculated effective densities can be used as direct inputs in respiratory dosimetry models as the MPPD model to predict deposition of nanoparticles in the human lungs. These predictions can be of high relevance for risk estimation or risk prioritization purposes, including Safe(r)-by-design strategies. For instance, differences in calculated depositions were found between the different processes for the same NM. Under RWC TiO_2_N coating process, the predicted daily deposited mass in the pulmonary region was 9.0 µg/day (7.3) for TiO_2_N (Appendix A), which is below the *no significant dose level* for particle overload, chronic inflammation and cell proliferation (300 µg/day) and tumor incidence (44 µg/day) in lungs suggested for TiO_2_ (respirable fraction including nano-TiO_2_) by Thompson et al. [41] for a lifetime exposure. This outcome suggest that processes are well controlled for TiO_2_ coating processes even at RWC where the worker spends the workday at the NF.

## 5. Conclusions

Process-specific emissions of TiO_2_N and AgHEC NPs generated during a spray coating manufacturing process were quantified. The data used are derived from a well-designed field exposure monitoring campaign in an industrial environment. The emission rates were utilized to identify safe CoU according to the regulatory guidance ECHA chapter R.14 by applying NIOSH recommended exposure limit values for nanosized TiO_2_ and Ag. The criteria for the CoU setting were selected as the 8-h 95th percentile of the lognormal distribution of exposures below 0.1×REL. The observed exposure scenario was considered as controlled even at the NF where the worker is not expected to spend more than 10% of the work time. Here, we demonstrated exposure scaling to full production under RWC conditions.

## Figures and Tables

**Figure 1 nanomaterials-12-00596-f001:**
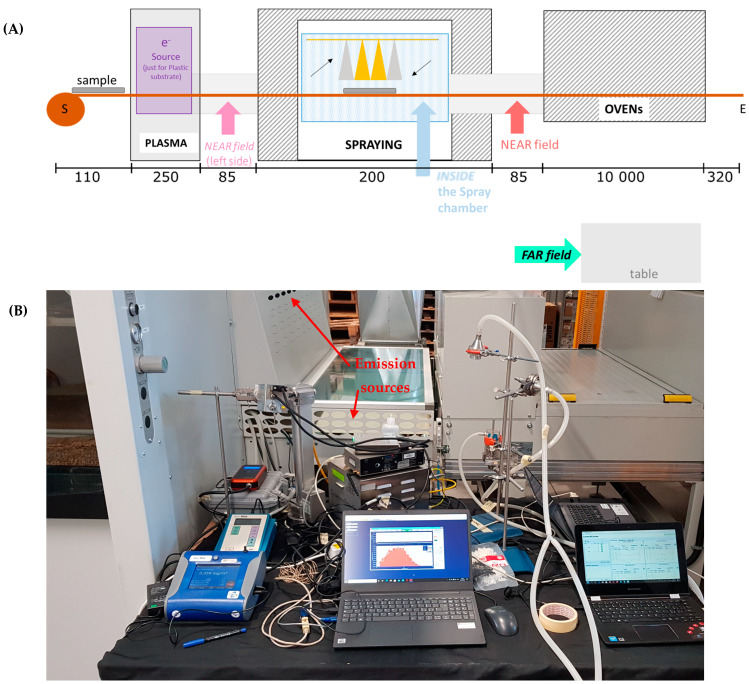
The coating system. (**A**) Schematic representation of the coating machine (not in scale; lengths are in cm), (**B**) NF measurement station at the spray chamber exit (red arrow in (**A**)). Stationary measurements were carried out from inside the spray chamber, NF and FF. Length of the coating system is ca. 22 m, the spray chamber width, length, and height are 3.5, 2.0, and 1.0 m, respectively, the conveyer belt width is 1.2 m and total with is 1.7 m and the belt speed can be adjusted from 0.1 to 1 m/min. Distance of NF instruments from the spray was ca. 1.5 m and the distance of FF from NF instruments was ca. 6 m.

**Figure 2 nanomaterials-12-00596-f002:**
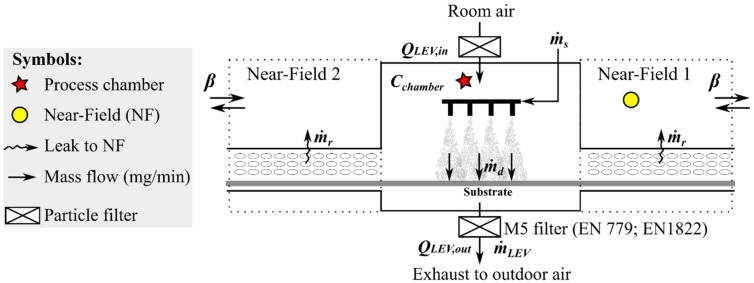
Schematic model of the spraying process. The spray chamber is assumed to be symmetric in leaking and mixing of pollutants at the conveyer belt inlet and outlet.

**Figure 3 nanomaterials-12-00596-f003:**
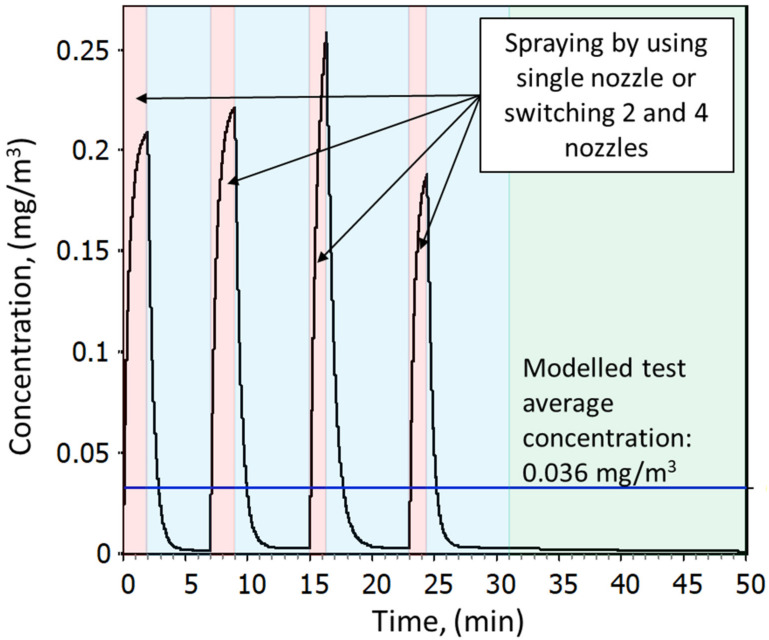
Example from a simulated test cycle (Test 1). Colored areas show the spraying times (red), pause between spray repetitions (blue), and the break for preparing a new test (green). Test cycle starts from the first spraying and ends to the start of next test. Blue line shows how test average concentration is calculated over the whole test cycle.

**Figure 4 nanomaterials-12-00596-f004:**
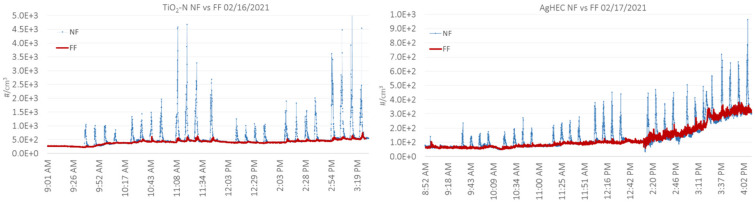
Particle number concentration measured by OPCs in the NF position (blue line) and FF (red line) for TiO_2_*N* suspension on the left and *AgHEC* suspension on the right.

**Figure 5 nanomaterials-12-00596-f005:**
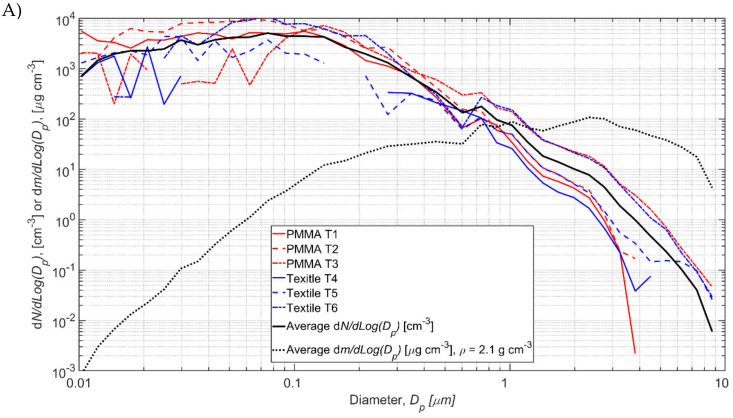
Process particle number size and mass size distributions for (**A**) TiO_2_N and (**B**) Ag experiments.

**Figure 6 nanomaterials-12-00596-f006:**
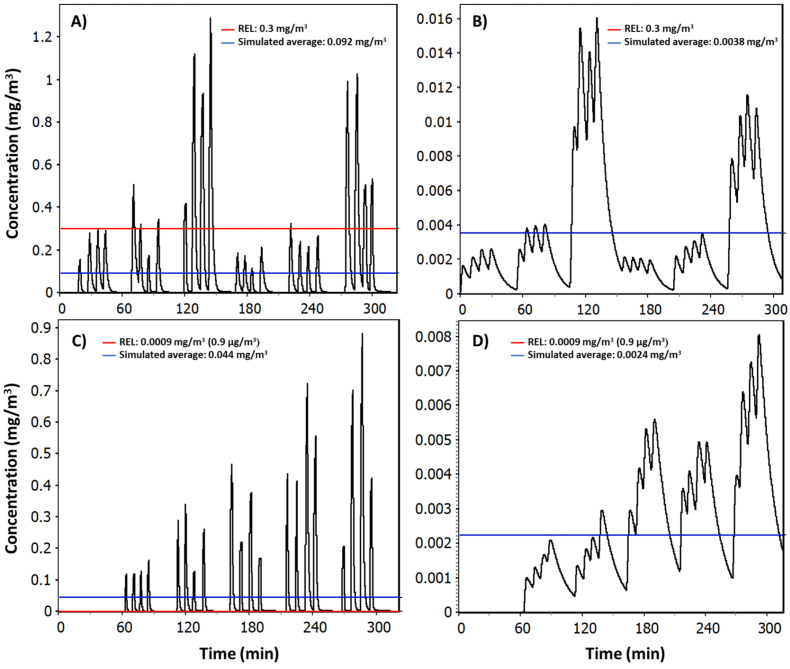
Simulated random day mass concentrations for TiO_2_N experiment at (**A**) NF and (**B**) FF and for AgHEC experiment (**C**) NF and (**D**) FF.

**Table 1 nanomaterials-12-00596-t001:** Parametrization of the work environment and tasks.

Work Environment (Same for All Tasks)		
Parameter	Value	Justification
Room volume, *V_room_* (m^3^)	Linear range: 1200 to 1470 m^3^	The room volume was 1470 m^3^, which was assumed to vary according to the storing conditions from 1200 to 1470 m^3^.
General ventilation air exchange rate, *AER* (1/h)	Linear range: 2.5 to 10 1/h	The room was naturally ventilated and by the LEV volume flow of 55.7 m^3^/min. Natural ventilation rate was not measured and it depends on climate and environmental. The natural ventilation was assumed to follow a triangular distribution with mode 2 1/h, minimum 0.5 1/h and maximum 8 1/h. The LEV increases the room ventilation rate by 2 1/h when turned on.
Flow rate between the spray chamber and NF, *Q_NF,SC_* (m^3^/min)	Constant: 0 m^3^/min	The LEV incoming and outgoing flow rates are in balance.
NF volume, *V_NF_* (m^3^)	Triangular distribution with: Min:1.0 m^3^ Mode: 3.0 m^3^ Max: 8.0 m^3^	The virtual NF volume was set as a cube covering the NF sources and instrument inlets (Figure 1). The cube side length was approximated with triangular distribution having a minimum 1.0 m, mode 1.5 m, and maximum 2.0 m. The NF volume was assumed to be closed from one side representing the spray chamber wall. The NF volume statistical simulation is presented in Appendix A.
Air mixing flow rate between NF and FF, β (m^3^/min)	Lognormal distribution with: GM: 5.58 m^3^/min GSD: 1.47	The NF volume is ventilated by a random air speed that enters the NF volume from one-half of the free surface area and exits the other one-half [34,35]: $\beta =\frac{1}{2}FSA\cdot s$, where *β* (m^3^/min) is the air mixing flow rate between NF and FF, *FSA* (m^2^) is the NF free surface area and *s* is the average random air velocity. The random air speed measured from ca. 2 m from the conveyer belt exit was up to 1.8 m/min (detection limit 1.8 m/min). The random air speed was assumed to follow a triangular distribution with mode 0.9 m/min, minimum 0.45 m/min, and maximum 1.8 m/min. The NF flow rate was statistical simulation and was approximated with a lognormal distribution with GM 5.588 m^3^/min and GSD 1.477 (Appendix A).

**Table 2 nanomaterials-12-00596-t002:** Materials, process parameters, particle number, mass concentrations, and emission rates in different tests. Test times are given in Appendix A, and parametrization of individual tests for emission rate simulation are shown in Appendix A.

Test No.	Material	No. of Nozzles	Substrate	*N_P,tot_*, [1/cm^3^]	*m_P,tot_*, [μg/m^3^]	Emission Rate *G*, [mg/min]	
						Total	NP
1	TiO_2_N, 1.0 wt.%	1	PMMA	6060	35.6	1.3	0.57
2	TiO_2_N, 1.0 wt.%	2	PMMA	8370	59.8	2.0	0.88
3	TiO_2_N, 1.0 wt.%	4	PMMA	3810	202.6	5.9	2.60
4	TiO_2_N, 1.0 wt.%	1	TEXTILE	520	25.1	0.9	0.40
5	TiO_2_N, 1.0 wt.%	2	TEXTILE	3040	49.5	1.6	0.70
6	TiO_2_N, 1.0 wt.%	4	TEXTILE	6670	181.5	4.5	1.98
1 to 6	TiO_2_N, 1.0 wt.%	Varying	Varying	4740	92.4	-	-
7	AgHEC, 0.01 wt.%	1	TEXTILE	38	-4.9^a^	N/A	N/A
8	AgHEC, 0.01 wt.%	2	TEXTILE	3400	17.9	0.8	0.009
9	AgHEC, 0.05 wt.%	1	TEXTILE	650	31.5	1.3	0.014
10	AgHEC, 0.05 wt.%	2	TEXTILE	2420	52.4	1.9	0.021
11	AgHEC, 0.1 wt.%	1	TEXTILE	1890	53.2	2.2	0.024
12	AgHEC, 0.1 wt.%	2	TEXTILE	3710	71.3	2.6	0.029
7 to 12	AgHEC, varying	Varying	TEXTILE	2020	36.9	-	-

**Table 3 nanomaterials-12-00596-t003:** Predicted total and NP exposure estimates in RWC during an 8-h work shift with two 3-h continuous spraying processes. Simulations were performed with Table 1 parametrization using precautionary emission rate with linear range as ${\dot{m}}_{r}$ ≤ *G* ≤ $2\cdot {\dot{m}}_{r}$. Weighted 95th percentile exposure was calculated as 10% of NF concentration 95th percentile and 90% of FF concentration 95th percentile. According to the ICP-MS analysis, total exposure levels during TiO_2_N and AgHEC spraying processes contains 44% and 1.1% TiO_2_ and Ag, respectively.

Test No.	Emission Rate Range *G*, [mg/min]	NF 95th Percentile, [μg/m^3^]		FF 95th Percentile *, [μg/m^3^]		Weighted 95th Percentile Exposure, [μg/m^3^]	
		Total	NP	Total	NP	Total	NP
1	1.3 to 2.6	466	205	44	19.4	86	38
2	2.0 to 4.0	710	312	68	29.9	132	58
3	5.9 to 11.8	2099	924	198	87.1	388	171
4	0.9 to 1.8	320	141	30	13.2	59	26
5	1.6 to 3.2	567	249	54	23.8	105	46
6	4.5 to 9.0	1601	704	152	66.9	297	131
7	N/A	N/A	N/A	N/A	N/A	N/A	N/A
8	0.8 to 1.6	287	3.2	26	0.3	52	0.6
9	1.3 to 2.6	462	5.1	60	0.7	100	1.2
10	1.9 to 3.8	676	7.4	64	0.7	125	1.5
11	2.2 to 4.4	787	8.7	74	0.8	145	1.8
12	2.6 to 5.2	961	10.6	88	1.0	175	2.1

* The FF concentration is doubled to estimate the NF2 concentrations dispersed to FF.

## Data Availability

The raw measurement data is available at [22].

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
