# Peer review of "Quantifying Emission Factors and Setting Conditions of Use According to ECHA Chapter R.14 for a Spray Process Designed for Nanocoatings—A Case Study"

_nanomaterials, 2022, doi:10.3390/nano12040596_

Round 1

Reviewer 1 Report

The authors characterized the process emissions from polymethyl methacrylate (PMMA) and textile substrates coated with TO2-N and Ag capped by hydroxyethylcellulose (Ag-HEC) NPs by using air spray guns. Emission rates were used to specify safe conditions of use according to ECHA chapter R.14 guidance by applying NIOSH recommended exposure limit values for nanosized TiO2 and Ag. The authors concluded that such emission rates can be used by Digital Twins to predict spray process risks under different operational conditions and optimize the manufacturing process.

The manuscript is well written and reports some important evidences about nanomaterials’ emission during spraying activities with consequences on occupational risk management and communication. I have some following comments, and I suggest publication after minor revision.

  1. In the Introduction section the authors reported few references about the state of the art on the subject which may support the aim and the relevance of the proposed study. The authors should include further references which report on the importance of safety issues incorporated in process optimization (i.e. safety by design or prevention through design approaches). Some examples are:
    1. Sanchez Jimenez et al., 2020 https://doi.org/10.1016/j.impact.2020.100267
    2. Boccuni et al., 2020 https://doi.org/10.1016/j.ssci.2020.104793
    3. Kraegeloh et al., 2018 doi:10.3390/nano8040239

Line 52: Please explain the reason why the use of Personal Protective Equipment (PPE) is recognized as “less effective risk control measures” by adding further references.

Lines 67-84: Please summarize and better clarify the scope of the study. In this section the authors report useless methodological information to be moved in the Methods section

  1. In the methods section the authors report that this work is based on the measurements described in paper by Del Secco et al. not already published. Even if the author allowed such paper as non published material, the present paper should be independent from the other one. For this reason please summarize in the methods section the following information related to the measurements:
    1. Producers’ data related to SMPS and two OPC;
    2. Detailed time schedule of measurements carried out on days 15-16-17 Feb;
    3. Workers’ position distance by the NF measurement point;
    4. Background measurements (time period, instrument position and calculated data including average value and standard deviation for real time data).

  1. Please report in the Methods section (or in supporting information) data on real-time instruments comparison (SMPS vs two OPCs) collected in parallel before or after the measurements, including also the OPCs intra-calibration results. If the authors do not performed an instrument comparison in this study, please discuss the issue in the Discussion section par. 4.2 “Model uncertainties”.

Author Response

The authors characterized the process emissions from polymethyl methacrylate (PMMA) and textile substrates coated with TO2-N and Ag capped by hydroxyethylcellulose (Ag-HEC) NPs by using air spray guns. Emission rates were used to specify safe conditions of use according to ECHA chapter R.14 guidance by applying NIOSH recommended exposure limit values for nanosized TiO2 and Ag. The authors concluded that such emission rates can be used by Digital Twins to predict spray process risks under different operational conditions and optimize the manufacturing process.

The manuscript is well written and reports some important evidences about nanomaterials’ emission during spraying activities with consequences on occupational risk management and communication. I have some following comments, and I suggest publication after minor revision.

  1. In the Introduction section the authors reported few references about the state of the art on the subject which may support the aim and the relevance of the proposed study. The authors should include further references which report on the importance of safety issues incorporated in process optimization (i.e. safety by design or prevention through design approaches). Some examples are:
    1. Sanchez Jimenez et al., 2020 https://doi.org/10.1016/j.impact.2020.100267
    2. Boccuni et al., 2020 https://doi.org/10.1016/j.ssci.2020.104793
    3. Kraegeloh et al., 2018 doi:10.3390/nano8040239

We thank the reviewer for the comment. A major adjustment was made to the introduction. We added a discussion including the references mentioned above.

Line 52: Please explain the reason why the use of Personal Protective Equipment (PPE) is recognized as “less effective risk control measures” by adding further references.

We thank the reviewer for the notice. PPEs are less preferred risk control measures which is addressed e.g. by Hirst et al. (2003) [2]. The sentence was removed and we added reference [2] at this sentence “Process emissions are a point-of-departure for subsequent exposure and health effects, thus, the quantification of them is foundational for effective risk control solutions [1], [2] and communication [3].”

[2] N. Hirst et al., ‘Occupational hygiene aspects of containment’, in Containment Systems, Elsevier, 2003, pp. 23–35. doi: 10.1016/B978-075067612-0/50004-X.

Lines 67-84: Please summarize and better clarify the scope of the study. In this section the authors report useless methodological information to be moved in the Methods section

We agree. We clarified better the study objectives and moved the methodological information to section 2.

In the methods section the authors report that this work is based on the measurements described in paper by Del Secco et al. not already published. Even if the author allowed such paper as non published material, the present paper should be independent from the other one.

We are happy to inform the reviewer that Del Secco et al. is now published at the same special issue where this manuscript is under review (B. Del Secco et al., ‘Particles Emission from an Industrial Spray Coating Process Using Nano-Materials’, Nanomaterials, vol. 12, no. 3, p. 313, Jan. 2022, doi: 10.3390/nano12030313.). While the paper is independent, we prefer to guide the readers also into the manuscript od Del Secco which we provide even more information on the same topic. We “over-cite” this manuscript because is the primary data source of our data analysis.

For this reason please summarize in the methods section the following information related to the measurements:

    1. Producers’ data related to SMPS and two OPC;

This is given in Del Secco et al. (2022)

    1. Detailed time schedule of measurements carried out on days 15-16-17 Feb;

This is given in Table S1, supplementary material 2, in addition to Del Secco et al. (2022).

    1. Workers’ position distance by the NF measurement point;

This work is based on stationary measurements performed in the NF and FF. In simulation, we assumed that the worker spends 10 % of the time during 8-h work shift in the in NF. We observed that the worker is mainly at the entrance of the plasma, at the ovens exit and follows the spray guns from ca 2 m from the NF.

    1. Background measurements (time period, instrument position and calculated data including average value and standard deviation for real time data).

Background concentrations are given in Table S2, S4.1 (number concentrations measured by OPS at FF) and S4.2 (calculated mass concentrations from the OPS at FF number concentrations), supplementary material. Del Secco et al. (2022) shows a sketch related to instrument locations.

Please report in the Methods section (or in supporting information) data on real-time instruments comparison (SMPS vs two OPCs) collected in parallel before or after the measurements, including also the OPCs intra-calibration results. If the authors do not performed an instrument comparison in this study, please discuss the issue in the Discussion section par. 4.2 “Model uncertainties”.

SMPS and OPC measures different metrics as discussed in Supplementary Material 3 and the measurement size range is different thus, we did not perform the comparison between SMPS and OPC. We added to section 2.2 “SMPS and both OPCs were calibrated by the manufacturer prior the campaign. OPC Near Field and OPC Far Field were also intercompared at CNR-ISAC by sampling in parallel aerosol particles at different concentrations. The OPC FF concentration was 1.1428 times the concentration measured by OPC NF in the particle concentration range from ca. 600 1/cm3 to 1600 1/cm3 with coefficient of determination R2 of 0.962.

             Below is also a figure showing the correlation (not presented in the manuscript).

Figure 1 – Laboratory intercomparison between OPCs. OPC (NF): OPC deployed at Near Field; OPC (FF): OPC deployed at Far Field.

Reviewer 2 Report

The work is interesting. But the organization of the paper should be improved, such as the sequence of the figure number is in a mess. Many spell typoses are found in the text, such as the subscript and formula. The format is no obey to the journal requirements.

Author Response

The work is interesting. But the organization of the paper should be improved, such as the sequence of the figure number is in a mess.

We corrected the figure numbering.

Many spell typoses are found in the text, such as the subscript and formula.

We corrected as well.

The format is no obey to the journal requirements.

We thank the reviewer for the notice. We changed the format accordingly.

Reviewer 3 Report

see attached

Author Response

Review of Nanomaterials 1531087

In general, this was a well-conducted and well described study of importance to worker health and safety as it pertains to nanoparticle exposures. The study results are well supported in most cases. A number of specific comments are provided below, most requesting additional statements to add clarity to a point made. Two, however, are of more importance. The first being the lack of stated objectives. This reduces the generalizability of this work, whereas the paper reads more like a case report of specific methods for a specific situation. It is likewise unclear whether there is anything performed here that is novel and pushes the science in this area. The other main comment is with regards to the addition of a dose assessment that is then not used, say, in a subsequent risk assessment. Both points are brought up below.

Specific Comments:

Introduction: Conclude with a brief explanation of the overall objectives of the study. Given the specific scenario in which the study was conducted, this statement should indicate the generalizable information obtained from this manuscript. Was this work conducted to ultimately relate how the described methods can be used by others in other scenarios, or was it conducted to best assess this scenario, or both? In other words, the “point” of this paper is not stated and is not obvious. Without describing the generalizability of this work, its overall impact is much diminished.

We agree. See our response from Reviewer 1 comment.

99: consider rewriting this grammatically incorrect sentence.

We agree, we changed the sentence to “Following process parameters that were expected to effect on the emissions were investigated:

109: change “red rectangle” with “red arrow”

This was changed.

138+: Methods to combine SMPS and OPC measurements was well described and sufficient.

No changes were made.

159+ in the steps add the word “number” or “mass” before “concentration” to distinguish between each.

We added the concentration specifications.

162: suggest that a reference be given here to section (2.4.2) to let the reader know “effective density” will be defined and discussed.

We added “Effective density take into account particle porosity and shape [29] and it can be calculated from mass and volume concentrations [30].”

206+ The description of “effective density” and the assumptions used is correct and adequate.

No changes were made.

225: The assumption stated here (“fully mixed”) is of major importance relative to the use of the model that relies on that assumption. Further explanation of the actual conditions of the room to reasonably ensure complete mixing should be given to support the assumption. Some of this justification is given in Table 1 but should be mentioned here first. For example, were fans added to enhance mixing and avoid dead space? And, a CO2 tracer study could have been performed to indicate the actual residence time relative to the ideal residence time to indicate the degree of mixing and incorporated into the model.

Uncontrolled air mixing is one big challenge in occupational mass flow analysis. However, controlled chamber studies are not possible to perform with large industrial machines. NF mixing is not possible because the particles would be carried out from the NF more uncontrolled way. FF concentration mixing could be enhanced by using mixing fans to mix concentration more homogenously but it is likely that it increases air mixing between NF and FF, β (m3/min). We tried to measure the concentration gradient in the NF by using DiskMini but it was not possible to perform with single instrument.

In section 4.2 Model uncertainties, we have written as “Major uncertainties in the model are related to air flow balance between the spray chamber and the NF (QNF,SC), how well the concentration measurements represent the NF average concentrations level, and the air mixing flow rate between NF and FF. Here the particles were assumed to diffuse from the spray chamber, i.e., QNF,SC = 0 m3/min, but if the LEV flows are not in balance it will change the emissions proportional to the QNF,SC. Concentration gradients in the NF volume can cause high under- or overestimation in emission rates. Concentration mapping requires simultaneous measurements with multiple instruments that were not available in this measurement campaign. Here, concentration measurements were expected to be representative for the mean NF concentration level because there were many small openings for particles to leak to the NF forming a local fugitive emission source rather than point source. Air mixing flow rate between NF and FF dilutes the NF concentrations. If dilution is overestimated, i.e., β is higher than true value, then the emission rate is overestimated. Here, the measured random air flow speed was up to 1.8 m/min (lower detection limit), which is below the typical random air speed at schools and offices of 2.5 m/min [53].

283: Using the MPPD model is an adequate method for estimating lung deposited does. The breathing parameter “3300 ml” should be explained. Is this the minute volume? And, it should be justified based on estimates of breathing rates for workers or the general population. See, for example, Janssen et al. Journal-International Society For Respiratory Protection. 2005;22(3/4):122.

We thank the reviewer for the comment. We made the following addition:

  • Breathing parameters for adult males were selected to reflect moderate worker activity: 3300 ml for functional residual capacity, 50 ml for extrathoracic volume, 20 breaths/minute for breathing frequency, and 1100 ml for tidal volume. This corresponds to inhalation rate of 1.36 m3/h which is between light (0.6 m3/h) to moderate (1.7 m3/h) activity for a 81 kg male between ages 35 and 64 [39]

It is also unclear, at this point, why an estimate of dose is needed when concentration is the important parameter as stated in the abstract. Some explanation should be given here. And, the results are provided in 3.10 but, without comparison values, these results are somewhat meaningless. For example, the MPPD model could be used to estimate the dose at the REL for comparison. (and see comment for line 648)

We added “Deposited dose of TiO2-N particles was compared with the no significant dose level of 300 µg/day for particle overload, chronic inflammation and cell proliferation and 44 µg/day for tumour incidence [40].

345: It is expected to report the GM and GSD of the distributions. In this case, an average for all trials would be sufficient.

Figure 5 shows the particle number size distributions and the values are given in Table S3.1, supplementary material. For Table S3.1, we calculated total number concentration, geometric mean diameter (GMD) and geometric standard deviation (GSD) by replacing negative values with zeros (Table R1). This method is not scientifically sounding, but the error here is not significant for number concentrations based on the difference between total number concentrations. Table R1 shows the GMD and GSD. Figure R1 shows that single mode log-normal fittings do not reproduce very well the measured distribution (note log-log scale). We prefer to leave out log-normal simplifications.

Table R1. Log-normal fittings for Table S3.1 number size distributions where negative values are replaced with zeros.

Test  No.

Material

Total number concentration

Geometric mean diameter

Geometric standard deviation

1

TiO2-N

6055

50

2.6

2

TiO2-N

8899

58

2.3

3

TiO2-N

3846

97

2.7

4

TiO2-N

795

33

3.5

5

TiO2-N

3075

43

2.4

6

TiO2-N

6867

83

2.1

7

Ag-HEC

1023

16

1.8

8

Ag-HEC

3525

55

1.6

9

Ag-HEC

774

34

2.6

10

Ag-HEC

2463

38

2.1

11

Ag-HEC

1908

45

2.5

12

Ag-HEC

3908

22

2.1

Figure R1. Number size distributions, dN/dLog(Dp), and log-normal distributions plotted with Table R1 values.

401: A brief explanation as to why 10 000 simulations should be given here by stating what was randomized in each run.

We added “The number of simulations was selected so that the results variation was insignificant when simulation was repeated by using the same scenario.”

405+: Again, there were no statements made as to the overall objective of this paper, but it would seem that validation of the modeling approach would be a principal objective. Here and the following statements are therefore among the most important of the paper: does the model adequately simulate reality?

We clarified this in the introduction as “The NF/FF model is validated in numerous studies and is accepted for regulatory occupational exposure assessment when applied properly [17]. Abattan et al. [16] showed that the NF/FF model predictive performance for solvents was within a factor of 0.3–3.7 times the measured concentrations with 93% of the values between 0.5 and 2. Similar results are obtained for various industrial processes [18]. Functioning models can be used to extrapolate observed concentrations or exposure levels in one operational condition to predict how the levels changes under different operational conditions. Such extrapolation is needed when production is scaled up or all operational conditions cannot be measured because of high variation of operational conditions e.g. between facilities [3]. Especially in nanotechnology industry scaling up the exposure measurements is needed because many of the processes are not yet adopted to full production scale and full-scale exposure measurements may not be feasible to perform.”

Concentrations are given (409-10) but it is not clear if they are modeled or actual.

We added “measured”

Exceptionally good ratios are given in line 412 without the evidence (here in the main document where it belongs) to support them. A table and/or figure showing simulations overlayed on measured values should be provided to support this important point of this work.

We added the evidence and explanation why the reproducibility was high “Emission rates were adjusted so that it reproduced the test specific mass concentrations (Table S5.1, Supplementary Material 2). Test specific emission rates were used to reproduce the full day NF measurements (Text S3 and S5, Supplementary Material 1). The ratio of the measured mass concentration mP,PM and the respective modelled mass concentration was 98% for the TiO2-N experiment and 94% for the Ag-HEC experiment i.e., the modelled concentration was slightly overestimated in both cases.”

We hope this also clarifies that this is not a validation study. A validation is to e.g., measure RWC concentrations in NF/FF and then see if the model actually can reproduce the exposure levels. This would show if the relevant exposure determinants are identified correctly and their magnitude are sufficiently accurately assigned.

521: “It was shown that the NF particle number concentration did not correlate well (correlation coefficients 522 are not calculated) with process parameters.” Where this information is “shown” is not obvious especially without r values.

The NF particle number concentration did not increase systematically with increasing number of nozzles for TiO2-N experiments or with increasing Ag-HEC concentration of the suspension for Ag-HEC experiments (Table 2; Figures S8.1 and S8.2, Supplementary Material 2). In mass concentration, there was a systematic increase in NF mass concentration with process parameters increasing the spraying process NP mass flow (Table 2; Figures S8.3 and S8.4, Supplementary Material 2).

604: “These significantly from the coefficient used by the OPC” Add “vary” before “significantly”?

This was added.

648: “These predictions can be of high relevance for risk estimation” Agreed, but that “risk estimation” was not performed here. So, again, the inclusion of a dose estimate here was purely “academic” and did not support any other aspect of this study, most notably a risk assessment. The authors should consider removing this aspect of the paper.

We compared TiO2 daily deposited dose in pulmonary region and compared for the limit values derived by Thompson et al. [4]. We modified the sentence to “Under RWC TiO2-N coating process the predicted daily deposited mass in the pulmonary region was 9.0 µg/day (7.3) for TiO2-N (Table S9.2, Supplementary Material 2), which is below the no significant dose level (NSRL) for particle overload, chronic inflammation and cell proliferation (300 µg/day) and tumor incidence (44 µg/day) in lungs suggested for TiO2 (respirable fraction including nano-TiO2) by Thompson et al. [40] for a lifetime exposure.” We consider that this is useful supporting information which we would like to include in the manuscript even though it is rough estimate and it consider only TiO2 and not Ag.

Conclusions: It is preferred that the statements made in the conclusion be restricted to the findings of the research described. Additional comments should be moved to the Discussion. That is true for most that is said after the sentence beginning on line 664.

We agree. Sentences were removed as proposed and replaced with a sentence “Here we demonstrated exposure scaling to full production under reasonable worst-case conditions.

Round 2

Reviewer 2 Report

the manuscript is well revised.